# Demolition of Existing Buildings in Urban Renewal Projects: A Decision Support System in the China Context

**Kexi Xu [1,2,\*], Geoffrey Qiping Shen [2,\*], Guiwen Liu [3] and Igor Martek [4]** 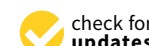

[1] School of Public Administration, Zhejiang University of Finance and Economics, Hangzhou 310018, China
[2] Department of Building and Real Estate, the Hong Kong Polytechnic University, Kowloon 999077, Hong Kong, China
[3] School of Construction Management and Real Estate, Chongqing University, Chongqing 400044, China; gwliu@cqu.edu.cn
[4] School of Architecture and Built Environment, Deakin University, Geelong, VIC 3220, Australia; igormartek@yahoo.com
\* Correspondence: xkxzj2017@zufe.edu.cn (K.X.); geoffrey.shen@polyu.edu.hk (G.Q.S.)

**Abstract:** Much of the rapid urbanization of China's cities has occurred at the expense of the existing urban fabric. Across the nation, whole city blocks have been replaced with new structures, requiring large numbers of buildings to be demolished while still serviceable. This curtailed lifespan of existing buildings not only comes with an economic cost, but results in loss of urban culture, wastes resources, degrades the environment, exacerbates pollution, and inflames social conflict and instability. For the purpose of evaluating the merits of building demolition, this study develops a decision support system (DSS) for building demolition in the China context from the perspective of sustainable urban renewal. The indicators of this system cover economic, social, environmental, and institutional aspects of sustainable development. Meanwhile, both the individual characteristics of buildings and the external or extrinsic indicators at the neighborhood, local, or city level are taken into account. Based on exploratory factor analysis (EFA) and confirmatory factor analysis (CFA), 24 critical indicators containing qualitative and quantitative factors are identified. These indicators are classified into six parameters: (1) service performance; (2) economic impact; (3) social identity; (4) local development; (5) building location; and (6) building safety. Empirical results reveal considerations of local development to be of greatest significance with the value of standardized factor loading standing at 0.911, followed by service performance (loading = 0.870) and building location (loading = 0.863), with social identity (loading = 0.236) ranking substantially lower. The findings contribute to the practice of urban renewal and, in particular, provide practical guidance to the building demolition decision-making process.

**Keywords:** building demolition; decision support system (DDS); confirmatory factor analysis (CFA); urban renewal; sustainable development

## 1. Introduction

Urban renewal significantly contributing to sustainable development has increasingly become a public profile within the urban policy agenda in both developed and developing countries [1,2]. As a typical developing country, China has undergone intense urbanization since China's reform and opening-up, beginning in 1978. By the end of 2017, the urbanization rate in China stood at 58.5% [3]. Prior to the 1980s, construction projects were government procured, being of extremely poor quality, yet overpriced [4]. Indeed, housing was frugal and was supplied in response to China's welfare-oriented

housing policy without consideration given to a real estate market, which did not emerge until the late 1990s. Thus, the majority of housing and infrastructure built before the 1990s is no longer fit for purpose and needs to be renewed. Moreover, China's rapid industrial modernization efforts have accelerated the need for extensive urban spatial adjustment and urban land use re-evaluation. Many cities are currently suffering from chronic land shortage combined with severe deterioration of the city centers. For example, in the city of Shenzhen, the so-called "three olds" of the city center—old town, old village, and old factory—remain unfit for contemporary use, yet cover some 240 km$^2$, or 5.5 times the area of the remaining available unused land in the city [5]. The experience of Shenzhen is typical, and highlights the point that urban renewal offers a resolution to the contradiction between urban development and China's land scarcity.

In recent years, extensive urban renewal projects throughout China have resulted in the large-scale demolition of existing buildings. According to a report by CABR (China Academy of Building Research) [6], the total area of demolished urban buildings reached 3 billion m$^2$ over the 11th National Five-Year Plan period (2006–2010), with the ratio of demolished buildings to newly constructed buildings approaching one-quarter, at 23%. During the 12th Five-Year Plan period, about 130 million m$^2$ of residential buildings was demolished in 2011 alone. By 2013, the total floor area of housing demolished rocketed to a staggering 400 million m$^2$. Certainly, demolition and redevelopment is a sure means to root out substandard buildings in order to replace them with new ones which meet construction standards and functional requirements. Moreover, redevelopment serves to eradicate out-of-date infrastructure, irrational utilization of land, and environmental hazards, while making it possible to more fully utilize urban land in ways that limit damage to the urban natural environment [7]. Nevertheless, the wholesale demolition of existing buildings without a re-evaluation of their potential for refurbishment, upgrading, or modification radically shortens the lifespan of buildings. Liu et al. [6] pointed out that the average service lifespan of demolished buildings in China is 34 years. This is both much shorter than the designed lifespan and shorter than the international level [8]. This shortening of building lifespan comes with great costs in the loss of urban culture, waste of resources, aggravation of environmental pollution, and emergence of social conflict and instability [9].

The continuing practice of the widespread demolition of existing buildings to facilitate the process of urban renewal has aroused intense public debate and widespread academic attention around the world, since urban renewal policies tend to emphasize sustainable development. Consequently, literature on sustainable urban renewal is now abundant. The concept of sustainable development is complex, while the opinion that it emphasizes the balance among the development of economy, society, and environment has been widely accepted in urban renewal [2]. Thus, most of the studies have tried to develop a decision support system (DDS) for urban renewal from specific aspects or a synthetic perspective of sustainable development [10–16]. Although previous studies provide numerous sets of indicators and frameworks collectively, due to a lack of verification, the range of indicators and framework structures in each remains distinct and different. Additionally, given the subjectivity of earlier research approaches, there is no agreement on the relative importance of the crucial indicators identified. Expert surveys are certainly helpful in the development of a decision support system (DDS) but bear the risk of generating unreasonably subjective results [17]. There is also the additional risk that quantitative indicators receive more attention while equally important qualitative factors are ignored. As an essential dimension of the majority of sustainable development assessment indices [18], such as Consultative Group on Sustainable Development Indicators, Environmental Sustainability Index, the political or institutional capacity (e.g., freedom to debate, private sector responsiveness) is almost overlooked in existing DDSs of urban renewal. Zheng et al. [2] warned that both quantitative and qualitative indicators must be regarded equally when making decisions on matters of urban renewal. On the other hand, the sustainability of building development is a crucial determinant of urban sustainability generally. Fundamental to this is city councils' long-term management of their portfolios of building stock [19]. Some studies have visited the question of how to balance the redevelopment and the renovation of the existing buildings based on the building itself [20–22].

Nonetheless, the majority of researchers paid more attention to local or area-based factors regarding urban renewal [2,10]. It is vital to take both external and internal factors into account for making decisions on urban renewal projects, since remarkable differences may exist among different buildings that are located in the same area or site, especially for historical areas or commercial centers.

Therefore, in areas undergoing urban renewal, a verified decision support system (DSS) covering the most recognized aspects (economy, society, environment, and institution) of sustainable development is essential to identify those buildings fit for demolition and those which ought to be preserved. This paper offers such a system with a multiscale consideration. First, 30 initial indicators were identified and collated through literature review and affirmed by expert interviews. Secondly, a questionnaire survey was conducted across six stakeholder groups nationwide to investigate these respondents' views of the 30 indicators. Thirdly, the collected data were processed and modelled using exploratory factor analysis (EFA) and confirmatory factor analysis (CFA). Twenty-four critical indicators and six constructs were derived, with the relative significance of each indicator quantified. Finally, a three-scale (building, neighborhood, and local characteristics) DSS was developed based on the analysis, with its significant features presented in this paper, along with a discussion of corroborations and differences with previous findings. The outcome of this study is submitted as a decision support system that may limit the unnecessary demolition of existing buildings while contributing to the optimization of sustainable urban renewal.

## 2. Literature Review

### 2.1. Criteria for Evaluating Building Serviceability

Various index systems for assessing the state of building conditions have been developed in previous studies, including for the buildings in U.K., USA, Japan, Singapore, and China. The English Housing Survey (EHS) [23] is an ongoing national survey of housing in the U.K., commissioned by the Department for Communities and Local Government. The EHS consists of both an assessment of the physical attributes of dwellings and an appraisal of the dwellings' market value. Physical attributes are evaluated in terms of both internal indicators, such as structural type, room usage, and maintenance condition, and external indicators, such as neighborhood quality, shared facilities, and access to common areas [23]. The market value survey provides two valuations: market valuation of the property in its current condition, and valuation once necessary repairs are undertaken. The Housing Quality Indicators system (HQI) in the U.K. emphasizes housing quality rather than merely cost [24] and is used to assess both new and existing residential buildings. This is reflected in the selection of indicators, which include such things as location, landscaping, layout, sustainability, and suitability of use.

In the USA, the Housing Choice Voucher (HCV) program aims to provide decent, safe, sanitary, and affordable housing to low-income families. In conjunction with the HCV program, the Housing Quality Standards (HQS) utilizes 13 key indicators as a basic guide for assessing the serviceability of potential and existing HCV housing [25]. Similarly, the Japanese government enacted the Housing Quality Assurance Act (HQAA) as early as 1999 in an effort to assure housing quality. The nine indicators used in the index system can be grouped into three categories: 1) safety (structural safety, fire safety, and deterioration mitigation), 2) indoor comfort (thermal, air, light, and acoustic environment), and 3) service performance (facilities for the elderly and those with disability and maintenance management) [26].

The Building and Construction Authority of Singapore utilizes the Existing Building Retrofit Guide (EBRG) for the assessment and retrofitting of existing buildings. The guide describes two dimensions by which the quality of existing buildings is evaluated: building performances (e.g., energy consumption, flexibility of floor plate) and building conditions (e.g., mechanical systems, structural condition) [27]. China has also followed the examples set in other countries. In 2005, the "Technical standard for performance assessment of residential buildings" (TSPARB) was brought into effect

by the Ministry of Housing and Urban Rural Development. The standard proposes five indicators for assessing both potential and existing housing quality. These indicators are usage, environment, economy, safety, and durability [28].

A side-by-side comparison of the indicators used by various national systems to evaluate building serviceability reveals a wide range of parameters, including safety, service performance, convenience (public service and facilities), comfort, economy (energy and resource saving), etc. (see Table 1). Moreover, external or extrinsic indicators, such as location and neighborhood characteristics, feature prominently in countries such as the U.K. However, as comprehensive as these assessment systems may be, their obvious limitation is that they apply only to residential buildings. Such systems were not designed and cannot be used to evaluate non-residential building usage, such as of industrial or commercial buildings. The urban landscape features the full complement of building types, and renewal projects must be able to evaluate the suitability of all the buildings that constitute the urban fabric. Furthermore, these building evaluation systems lack lifecycle-based indicators [29], such as the economic and social benefits that may be afforded by a building over time. A further weakness is that such systems focus on current static conditions rather than providing guidance in decision-making under dynamic or evolving conditions. Singapore's EBRG alone does offer a guide for decision-making on existing building retrofitting, but the indicators are far from comprehensive.

**Table 1.** Comparison of national index systems used to assess building conditions.

| Building Assessment Index<br>Year of Introduction | | HQI<br>2008 | EHS<br>2014 | HQS<br>2001 | HQAA<br>1999 | TSPARB<br>2005 | EBRG<br>2010 |
|---|---|---|---|---|---|---|---|
| **Country of Origin** | | U.K. | | USA | Japan | China | Singapore |
| Building types | New building | √ | | √ | √ | √ | |
| | Existing building | | √ | | | | √ |
| Intrinsic indicators | Structural safety | √ | √ | √ | √ | √ | √ |
| | Fire safety | √ | √ | √ | √ | √ | √ |
| | Indoor environment (indoor comfort, air quality) | √ | √ | √ | √ | √ | √ |
| | Usage convenience (equipment system, facilities) | √ | √ | √ | √ | √ | |
| | Unit interior (unit layout, unit size) | √ | √ | | | √ | |
| | Building scale | | | √ | | √ | |
| | Energy and resource consumption | √ | | | | √ | |
| | Architectural quality | √ | | | | √ | |
| | Distinctive character | √ | | | | √ | |
| Extrinsic indicators | Building value (location, condition of dwelling) | | | √ | | | |
| | Market demand of existing building (number of vacant units) | | | √ | | | |
| | Nature disaster impact (flood, mudslide) | | | | √ | | |
| | Environmental safety | √ | | | √ | √ | |
| | Environmental context (visual impact, landscaping) | √ | √ | | | √ | |
| | Integration with existing buildings, landscape, and topography | √ | | | | √ | |
| | Transport availability | √ | | | | √ | |
| | Access to public facilities | √ | √ | | | √ | |
| | Provision of local community needs | √ | | | | | |
| | Integration with surrounding development | √ | | | | | |

## 2.2. Decision Support Indicators for Sustainable Urban Renewal

Moving from criteria to assess specific residential buildings to the macro consideration of systems established to support decisions on sustainable urban renewal, we found that a number of researchers have developed indicator-based frameworks. Goodchild [30] was among the first to develop evaluation systems to support decisions on sustainable urban renewal. He observed that a building's condition combined with the degree of project urgency directly influenced the choice of renewal strategy. Juan et al. [21] refined the observation and found six criteria, based on a literature review, relevant to applications for renovation: safety, usage, convenience, comfort, utility, and health. Instead of focusing on the physical conditions of a building, more researchers developed decision-making frameworks for urban renewal projects based on multiple dimensions of sustainability. Ng et al. [31] noted that the social, economic, environmental, and traffic effects imposed by an urban renewal project become crucial components in the renewal agenda. Yau and Ho [20] developed a multicriteria decision support framework, able to facilitate the evaluation of urban renewal schemes, using 16 criteria grouped under four categories: environmental, physical, and social conditions and economic benefit. Langston and Wong [32] explored the relationship between the financial, environmental, and social parameters associated with building adaptive reuse and used these to develop an adaptive reuse potential (ARP) model for existing buildings.

In terms of specific dimensions of sustainable development, the environmental and social impacts deserve more attention in urban renewal [33,34]. Itard and Klunder [7] developed a comparison between the external environmental impacts on renovation and the reconstruction of existing buildings based on the method of life cycle evaluation. They pointed out that environmental indictors are significant to the decision-making process in urban renewal. Saldaña-Márquez et al. [35] found significant deficiencies in materials, energy efficiency, and indoor environmental quality resulting in low qualifications of social housing in Mexico and indicated that developing an evaluation model which gives priority to urban environmental impact might be helpful for achieving the sustainability of social housing. With respect to the social dimension, Yu et al. [36] developed an assessment model for the social sustainability of housing demolition in Shanghai based on a two-wave questionnaire survey. The results showed that health and safety, social equality, and adherence to the law are the most critical indicators influencing social sustainability.

Apart from an assessment of the conditions of the building itself and the surroundings, there remains the question of how the building is to be renewed and renovated. There will, of course, be any number of renewal alternatives, with the choice dependent on multiple considerations [21]. Kaklauskas et al. [37] identified the essential indicators—cost of renovation, annual fuel expenditure, pay-back period, harmfulness to health of the materials used, aesthetics, maintenance properties, functionality, comfort, sound insulation and longevity, etc.—for the efficiency evaluation of a building's renovation. Juan et al. [38] developed an integrated decision support system which considers a balance of renovation costs, improved building quality, and environmental impacts as the basis to assess current conditions and to recommend an optimal set of sustainable renovation strategies. For existing office buildings, Brandt and Rasmussen [39] built a TOBUS method able to suggest building upgrading solutions in response to an appraisal of current building conditions (such as degree of functional degradation, service level, and indoor air quality), amount of essential work for renovation, and costs.

Certainly, external or extrinsic indicators at the neighborhood, city, or local level, as well as internal or intrinsic indicators at the level of the building itself, are vital to the decision-making for urban renewal. Zheng et al. [14] established a comprehensive evaluation system for urban renewal by adopting a method of multiscale decision-making. According to Liu et al. [9], building characteristics, location characteristics, neighborhood characteristics, economic variables, and political variables are the key determinants of building demolition with respect to urban renewal in China. Moreover, there are a number of studies which focus on the assessment of sustainable urban renewal at the local or site-based level [40–42]. Indicators related to the key characteristics of local development that have been identified, including building location, real estate market, land use, transportation and mobility,

population, and the economy. As factors informing the decision-making process with respect to urban renewal projects, all these factors have been documented in previous studies.

## 3. Methodology

The data for this research were collected by using a combination of literature review, semi-structured expert interview, and questionnaire survey. The literature review and interviews of experts in urban renewal were employed to generate the preliminary list of indicators of the DSS for building demolition. Once the list was established, the questionnaire survey was conducted to evaluate the adequacy and importance of each potential indicator among the various classes of stakeholders related to building demolition. Using exploratory factor analysis (EFA) and confirmatory factor analysis (CFA), the data were processed and modeled to discriminate the critical indicators and quantify the relative importance of each indicator. Finally, an indicator-based DSS with stable constructs for building demolition in urban renewal projects was developed.

### 3.1. Preliminary List of Indicators

Through the adoption and consolidation of previous research findings, a comprehensive initial list was collated from existing literature [9–16,21,33,38,43–47]. The list was then refined through a series of interviews with twelve experts on urban renewal in July 2017. These experts were located in Shenzhen and Chongqing, and comprised three officials from the Urban Renewal Authority of Shenzhen, three from the Bureau of Housing and Urban–Rural Development of Chongqing, two from the Bureau of Urban Planning of Shenzhen, two senior executives from construction project decision-making consultancies, and two academics from universities. As an outcome of these interviews, 30 decision indicators were ultimately selected. These are summarized in Table 2.

The 30 indicators can be divided into two groups of 15 indicators each (15 intrinsic or internal indicators (X1–X7, X12–X13, X16, X18–X22) and 15 extrinsic or external indicators (X8–X11, X14–X15, X17, X23–X30)), covering aspects of the building's physical condition, economic impact, social identity, environmental sustainability, and institutional capacity. The building's physical condition refers to the safety performance and the service performance of a building, such as structural safety (X1), indoor comfort (X3), and usage convenience (X4). The indicators underlying economic impact can be grouped into two clusters: 1) economic assessment of building itself, e.g., life-cycle cost (X12), building value (X13), and rate of return on investment (ROI) (X16); and 2) development of local economy, e.g., land expectation value (X14), infrastructural investment (X28), and industrial adjustment (X30). Social identity is influenced by the factor of culture. Therefore, such cultural indicators, e.g., historic cityscape (X19) and architectural features (X22), are classified under the dimension of society. Environmental sustainability reveals the resource consumption and the environmental impact regarding different urban renewal scenarios, including the life-cycle consumption of resources and energy (X7), environmental harmony (X11), etc. In addition, both the political indicators and the indicators of stakeholder contentment (public participation), including household wishes (X18), consistency of local planning (X26), and sustainability of urban planning (X27), etc., are deemed the indicators for the assessment of institutional capacity in urban renewal decision-making.

**Table 2.** Key decision support indicator candidates for building demolition in urban renewal projects.

| Indicators | Illustration | Code | Key References |
|---|---|---|---|
| Structural safety | Degree of building damage (construction quality, foundation, level of load, earthquake proofing) | X1 | |
| Fire safety | Fireproof endurance rating, fire facilities, structural fire protection, evacuation facilities | X2 | |
| Indoor comfort | Indoor comfort level (sound environment, light and thermal environment) and air quality | X3 | [7,12,23–29,37] |
| Usage convenience | Installations and service levels of facilities, including hydropower, pipelines, elevators, accessibility, and aging facilities | X4 | |
| Unit interior | Spatial function, unit layout, unit size, suitability | X5 | |
| Building scale | Total floor area of the existing building | X6 | |
| Life-cycle consumption of resources and energy | Energy and resource (water, land, materials) savings of an existing building throughout its life-cycle (including construction, maintenance, renovation, and demolition) | X7 | |
| Natural disasters | The frequency of natural disasters in the urban renewal site | X8 | |
| Environmental safety | Distance to pollution and other dangerous sources, such as sewage treatment plants, garbage disposal plants, hazardous facilities, etc. | X9 | [9,10,21] |
| Amenities condition | Distances to natural amenities (rivers, lakes, mountains, etc.) and human amenities (gardens, schools, etc.) | X10 | |
| Environmental harmony | Whether the existing building will harm the surrounding biological environment; whether the surrounding environment can meet the requirements of relevant codes | X11 | |
| Life-cycle cost | Total cost of a building over its life-cycle, including the cost of construction, maintenance, renovation, and demolition (including land acquisition and resettlement costs) | X12 | [29,48] |
| Building value | Comparative rents between different existing buildings with same function in the same neighborhood | X13 | [23] |
| Land value | Expected value of land, including the existing building, over a set planning period | X14 | |
| Market demand of the existing building | Vacancy rate of existing buildings in the neighborhood (this indicator represents the acceptance of existing buildings as some old buildings with low rent are popular in low-income populations) | X15 | [43] [45] |
| Rate of return on investment (ROI) | Gaps of ROI among different kinds of urban renewal scenarios (redevelopment, renovation, maintenance, etc.) | X16 | [47] |
| Households' wishes | Households' wishes regarding the demolition of existing buildings | X17 | Expert interview |
| Continuity of history | Value obtained as a carrier of historical data, historical events, historical figures, etc. | X18 | [8–10,15] |
| Historic cityscape | Value obtained as a carrier of social characteristics over a specific period of urban development | X19 | |
| Architectural aesthetics | Value obtained as a carrier of a unique example of a scarce building type | X20 | |
| Construction technologies | Value obtained from a construction method or engineering technology representative of a specific time | X21 | Expert interview |
| Architectural features | Distance from the existing building to a historic district or heritage building; value obtained from the coordination degree of architectural features between the existing building and surroundings (historic block/district, heritage building) | X22 | |
| Traffic availability | The walking distance from the existing building to nearest public transport (bus station, railway station, etc.) | X23 | [9,13,20–25,30] |
| Commercial location | Distance from the existing building to CBD/commercial center | X24 | |
| Availability of public facilities | The completeness and convenience of public services and facilities in the neighborhood and local level | X25 | |
| Consistency of local planning | Whether the condition of the existing building can meet the requirements of related planning, such as local development planning, land use planning/Whether the development potential of the site has been substantially utilized or not | X26 | [18,40] Expert interview |
| Sustainability of urban planning | The reasonability of amending urban planning | X27 | |
| Infrastructural investment | The investment of infrastructure in the urban renewal site, such as roads, bridges, etc. | X28 | [36,38] |
| Demand for urban building space | Change rate of urban population | X29 | [36,38] |
| Industrial structure adjustment | The variation rate of proportion of industries in the urban renewal (the degree to which adjustment of industrial structure will lead to the displacement of urban spatial structure) | X30 | [7,49] |

### 3.2. Questionnaire Design

In order to collect stakeholders' opinions on these indicators, a two-part questionnaire was developed. The first part was designed to collect the demographic information of respondents, e.g., stakeholder type and years of experience. In the second part, respondents were asked to assess the relative significance of the 30 indicators on a 5-point Likert scale, from their own perspective, by scoring each indicator from 1 ("least significant") to 5 ("most significant"), with 3 being neutral. Various classes of stakeholders in the building demolition decision-making process have previously been identified [12,50,51]. These include government officials, developers and investors, building constructors, scholars, and specialist experts (such as designers and project managers), as well as relocated householders and other invested social groups (such as neighborhood residents). In order to reflect the views of all these classes of stakeholders properly, the survey was conducted across this full range of people.

### 3.3. Data Collection

The full survey was conducted nationwide over a two-month period from September to November 2017. Invitation letters were sent to the identified respondents, with the questionnaire sent via e-mail or post thereafter. Ultimately, a total of 500 questionnaires were dispatched, with 276 valid responses returned, representing a response rate of 55.2%. Of those respondents who replied, 44% had more than 5 years' experience in urban renewal and building demolition, while 25% had acted in the capacity of decision-maker concerning building demolition in urban renewal projects. The distribution of the various types of respondents is summarized in Table 3.

**Table 3.** Distribution of questionnaire respondents by stakeholder type.

| Stakeholder | Frequency | Percentage |
| --- | --- | --- |
| 1 - Government official | 21 | 7.6% |
| 2 - Developer or investor | 89 | 32.3% |
| 3 - Constructor | 105 | 38.1% |
| 4 - Relocated householder | 14 | 5.0% |
| 5 - Scholar or expert | 32 | 11.6% |
| 6 - Social group or affected public | 15 | 5.4% |
| Total | 276 | 100.0% |

### 3.4. Data Analysis

Exploratory Factor Analysis (EFA) is traditionally used to explore the possible underlying factor structure of a set of observed variables without imposing a preconceived structure on the outcome [51]. In this study, EFA was utilized as a generator of hypotheses of the factorial structure. Confirmatory Factor Analysis (CFA) is a hypothesis testing technique that statistically tests the fit of a predetermined model. Thus, performing the CFA of a model which was developed through EFA is a viable strategy for theory development and analysis [51]. Generally, structural equation modelling (SEM) software is adopted to perform CFA, since CFA is deemed a special application of SEM [52]. In the context of SEM, CFA is known as the measurement model because it is conducted to identify how latent variables (unobserved dimensions) are measured by their underlying observed variables (indicators) [53]. The equation of the measurement model used in this study, as derived by Gerbing et al. [54], is as follows:

$$y = \Lambda_y \eta + \varepsilon. \tag{1}$$

In the equation, $y$ denoted the observed indictors of $\eta$, $\eta$ denotes the latent variables, $\Lambda_y$ shows the factor loading of $y$, and $\varepsilon$ represents the measurement errors for $y$. In this study, a second-order CFA model of the building demolition DSS was assumed. The latent variables directly influence the valid observed indicators, while the unobserved dimensions are influenced by a higher-order dimension,

namely, the DSS. Thus, the DSS does not necessarily explain the observed indicators directly. It is recommended that the subject-to-item ratio for EFA should be at least 5:1, with the absolute acceptable sample size generally ranging from 50 to 400 [55]. Hair et al. [56] suggested that the minimum required amount of samples for adopting CFA should be between 100 and 150. Therefore, 150 samples were extracted randomly from the total of 276 samples for EFA, while the other 126 samples were used for CFA.

In this study, the EFA was run via SPSS 17.0 and the CFA was conducted using AMOS 20.0, from SEM software. On the basis of the principal components extracted through EFA, a default second-order CFA model was developed. As the default second-order model, all constructs (observed indicators and latent variables) were allowed to be correlated freely. Maximum Likelihood (MI) is widely used to achieve more accurate parameter values when the sample size is less than 2000 [57]. It was therefore adopted as the estimation method in processing the CFA. The model was refined according to tests of reliability and validity. Since the main contribution of this study is the validation of the newly developed DDS with CFA, only the final CFA model was selected for analysis [51].

## 4. Results

### 4.1. Consistency Analysis among Response Groups

In this study, since the data were collected from six different groups, an analysis of consistency of the relative significance of indicators among the six groups of respondents was necessary. Babin et al. [56] confirmed that the ANOVA test is applicable as a test for consistency of responses among different types of respondents. Where the *p*-value of the ANOVA test is lower than 0.05, it is safe to say that a significant variance exists, while a *p*-value of less than 0.01 indicates an extremely significant difference. However, due to a significant difference among different groups in the number of samples, it is essential to first do a "homogeneity test of variance" before conducting the ANOVA test. Applying the homogeneity test, all indicators except X17, X20, and X25 reached a *p*-value of 0.05 or greater, and were therefore eligible for the variance analysis test [56] (see Table 4). Table 5 shows the results of the ANOVA analysis. It can be seen that the indicators X7 and X22 have an extremely significant difference (*p*-value = 0.01) among different responding groups, while there is no significant variance among different categories for the other 25 indicators.

**Table 4.** Homogeneity test of variance.

| Indicator | Levene Value | df1 | df2 | Sig. | Indicator | Levene Value | df1 | df2 | Sig. |
|-----------|--------------|-----|-----|------|-----------|--------------|-----|-----|------|
| X1 | 1.53 | 5 | 270 | 0.18 | X16 | 0.39 | 5 | 270 | 0.85 |
| X2 | 2.15 | 5 | 270 | 0.06 | X17 | 3.02 | 5 | 270 | 0.01 |
| X3 | 1.00 | 5 | 270 | 0.42 | X18 | 2.26 | 5 | 270 | 0.05 |
| X4 | 1.50 | 5 | 270 | 0.19 | X19 | 0.81 | 5 | 270 | 0.55 |
| X5 | 1.32 | 5 | 270 | 0.26 | X20 | 3.29 | 5 | 270 | 0.01 |
| X6 | 1.05 | 5 | 270 | 0.39 | X21 | 0.89 | 5 | 270 | 0.49 |
| X7 | 0.92 | 5 | 270 | 0.47 | X22 | 1.58 | 5 | 270 | 0.17 |
| X8 | 0.58 | 5 | 270 | 0.72 | X23 | 0.61 | 5 | 270 | 0.69 |
| X9 | 1.21 | 5 | 270 | 0.30 | X24 | 0.49 | 5 | 270 | 0.79 |
| X10 | 1.69 | 5 | 270 | 0.14 | X25 | 2.61 | 5 | 270 | 0.03 |
| X11 | 1.66 | 5 | 270 | 0.14 | X26 | 0.56 | 5 | 270 | 0.73 |
| X12 | 0.23 | 5 | 270 | 0.95 | X27 | 1.42 | 5 | 270 | 0.22 |
| X13 | 0.98 | 5 | 270 | 0.43 | X28 | 1.20 | 5 | 270 | 0.31 |
| X14 | 0.29 | 5 | 270 | 0.92 | X29 | 2.04 | 5 | 270 | 0.07 |
| X15 | 0.97 | 5 | 270 | 0.44 | X30 | 0.66 | 5 | 270 | 0.65 |

**Table 5.** ANOVA test results.

| Indicator | Sig. | Indicator | Sig. | Indicator | Sig. |
|:---:|:---:|:---:|:---:|:---:|:---:|
| X1 | 0.61 | X10 | 0.02 | X21 | 0.02 |
| X2 | 0.28 | X11 | 0.04 | X22 | 0.01 |
| X3 | 0.08 | X12 | 0.87 | X23 | 0.20 |
| X4 | 0.02 | X13 | 0.28 | X24 | 0.32 |
| X5 | 0.05 | X14 | 0.19 | X26 | 0.91 |
| X6 | 0.03 | X15 | 0.90 | X27 | 0.46 |
| X7 | 0.01 | X16 | 0.56 | X28 | 0.22 |
| X8 | 0.17 | X18 | 0.21 | X29 | 0.59 |
| X9 | 0.18 | X19 | 0.28 | X30 | 0.18 |

### 4.2. Reliability Analysis

Cronbach's alpha ($\alpha$) is generally utilized to validate the reliability of responses. Values range from 0 to 1, where the larger the value of Cronbach's $\alpha$, the better the reliability of the internal consistency of the data. The minimum acceptable value of Cronbach's $\alpha$ is 0.6, while a value higher than 0.8 indicates excellent reliability [56]. The reliability computation for this study was conducted using SPSS 17.0. The Cronbach's $\alpha$ values of the total samples and the 150 extracted samples came in at 0.932 and 0.930, respectively. These results confirm that the questionnaire results are most reliable.

### 4.3. KMO and Bartlett's Tests

Further confirmations of the reliability of the data are available through the Kaiser–Meyer–Olkin measure of sampling adequacy, or the KMO test, and Bartlett's test of sphericity [58]. The value from the KMO test ranges from 0 to 1. The closer the KMO value is to 1, the more amenable the variables are to factor analysis [59]. Previous studies have suggested that a value over 0.7 meets minimal suitability for EFA, while a value higher than 0.8 represents strong suitability [60]. The KMO test results for the collected data achieved a value of 0.895, indicating that the 150 extracted samples are suitable for EFA. Moreover, the *p*-value of the Bartlett's test was 0.00, which is less than the significance level of 0.05, indicating a strong concomitant probability among samples [58] (see Table 6). Overall, these tests confirm that the collected data is wholly fit for analysis by the EFA method.

**Table 6.** Results of Kaiser–Meyer–Olkin (KMO) and Bartlett's tests.

| Test | Frequency | | Value |
|:---:|:---:|:---:|:---:|
| KMO test | 150 | | 0.895 |
| Bartlett's test | 150 | Approx. Chi-Square | 2,264.093 |
| | | Sig. | 0.00 |

### 4.4. Determination of Principal Components

Principal component analysis with varimax rotation was conducted on the 30 initial indicators for the DSS of building demolition. According to Comrey and Lee [61], the minimal factor loading of a variable with explanatory power is 0.32, while a factor loading greater than 0.45 indicates a moderate association, whereas a factor loading greater than 0.71 provides strong explanatory power. Consequently, three indicators found to have a factor loading of less than 0.45 were rejected during the factor analysis. These were building scale (X6), environmental harmony (X11), and households' wishes (X17). Based on the precedent of accepting eigenvalues higher than 1.0 and cumulative variance contribution rates greater than 60% [62], six principle components with 27 indicators were extracted (see Table 7). The percentage of variance extracted by the six factors was 66.3%.

**Table 7.** Principal factor extraction and varimax rotations of crucial indictors.

| Factors | Indicators | Factor Loading | | | | | | Communalities |
|---|---|---|---|---|---|---|---|---|
| | | 1 | 2 | 3 | 4 | 5 | 6 | |
| Service Performance (SP) | X4 Usage convenience (SP1) | 0.78 | | | | | | 0.75 |
| | X5 Unit interior (SP2) | 0.74 | | | | | | 0.68 |
| | X3 Indoor comfort (SP3) | 0.67 | | | | | | 0.65 |
| | X10 Amenities condition (SP4) | 0.64 | | | | | | 0.56 |
| | X7 Life-cycle consumption of resources and energy (SP5) | 0.49 | | | | | | 0.59 |
| Economic Impact (EI) | X16 Rate of return on investment (ROI) (EI1) | | 0.81 | | | | | 0.71 |
| | X14 Land expectation value (EI2) | | 0.80 | | | | | 0.70 |
| | X13 Building value (EI3) | | 0.73 | | | | | 0.66 |
| | X15 Market demand of existing building (EI4) | | 0.70 | | | | | 0.68 |
| | X12 Life-cycle cost (EI5) | | 0.57 | | | | | 0.51 |
| Social Identity (SI) | X20 Architectural art (SI1) | | | 0.84 | | | | 0.77 |
| | X19 Historic cityscape (SI2) | | | 0.74 | | | | 0.73 |
| | X18 Continuity of history (SI3) | | | 0.72 | | | | 0.60 |
| | X22 Architectural features (SI4) | | | 0.68 | | | | 0.66 |
| | X21 Construction technologies (SI5) | | | 0.67 | | | | 0.66 |
| Local Development (LD) | X29 Rigid demand of urban building space (LD1) | | | | 0.73 | | | 0.66 |
| | X30 Industrial structure adjustment (LD2) | | | | 0.70 | | | 0.63 |
| | X28 Infrastructural investment (LD3) | | | | 0.66 | | | 0.64 |
| | X27 Sustainability of urban planning (LD4) | | | | 0.55 | | | 0.68 |
| Building Location (BL) | X25 Availability of public facilities (BL1) | | | | | 0.76 | | 0.80 |
| | X24 Commercial location (BL2) | | | | | 0.73 | | 0.71 |
| | X23 Transportation availability (BL3) | | | | | 0.64 | | 0.67 |
| | X26 Consistency of local development (BL4) | | | | | 0.53 | | 0.63 |
| Building Safety (BS) | X1 Structural safety (BS1) | | | | | | 0.80 | 0.68 |
| | X9 Environmental safety (BS2) | | | | | | 0.61 | 0.65 |
| | X8 Natural disasters (BS3) | | | | | | 0.57 | 0.55 |
| | X2 Fire safety (BS4) | | | | | | 0.56 | 0.67 |
| Characteristic value | | 3.71 | 3.35 | 3.12 | 2.83 | 2.63 | 2.27 | - |
| Contribution rate/% | | 13.75 | 12.41 | 11.51 | 10.49 | 9.72 | 8.40 | - |
| Cumulative % | | 13.75 | 26.16 | 37.67 | 48.16 | 57.88 | 66.28 | - |

The six factors extracted from the 27 remaining indicators are service performance, economic impact, cultural value, local development, building location, and building safety. The factors of Service Performance (SP) and Building Safety (BS) involve nine indicators in total. Among them, four indictors, namely, amenities condition, life-cycle consumption of resources and energy, environmental safety, and natural disasters, are all related to the environment, whilst the others are regarded as indicators of building conditions. In terms of Economic Impact (EI), five indicators were identified, including two static indicators (building value, market demand on existing building) and three dynamic indicators (land expectation value, ROI, life-cycle cost). As for Social Identity (CV), five indicators were identified, related to building characteristics (e.g., architectural features and construction technologies) and local context (e.g., historic cityscape, continuity of history). Moreover, there are four indicators that can be said to emphasize the neighborhood characteristics of existing buildings, since the consistency of local development can be equated with political location. Consequently, this component contains these four indicators, and has been labelled Building Location (BL). Finally, the four remaining indicators, of demand for urban building space, industrial structure adjustment, infrastructural investment, and sustainability of urban planning, together demonstrate the importance of Local Development (LD) as an assessment dimension of the DSS of building demolition in urban renewal.

*4.5. Overall Model Fit Test*

An overall fit test aims to evaluate the difference between the correlation matrix of the observed indicators and the predicted correlation matrix of the model [63]. In this study, two absolute fit measures ($\chi^2$, root mean square error of approximation), two incremental fit measures (comparative fit index, Tucker-Lewis index), and three parsimonious fit measures (parsimony normed-fit index, parsimony goodness-of-fit index, $\chi^2/df$) were utilized [64]. The values of $\chi^2/df$, parsimony normed-fit index (PNFI), and parsimony goodness-of-fit index (PGFI) of the default model are acceptable. However, the overall fit of the default model indicates that this model ought to be modified since the values of root mean square error of approximation (RMSEA), comparative fit index (CFI), and Tucker-Lewis index (TLI) were below acceptable limits (see Table 8). In order to achieve an acceptable fit of the data, three indicators were discarded from the latent variables and seven covariant relationships were built among the indicators based on the assumptions of the SEM and theoretical model [65] (see Figure 1). Thus, with these adjustments giving a modified model, all three types of goodness of fit indices were satisfied, confirming that the modified model meets statistical standards of fit. This modified measurement model can be taken as a reliable expression of the sample data.

**Table 8.** Results of the overall fit test for the second-order Confirmatory Factor Analysis (CFA) model of the decision support system (DSS).

| Indices | Default Model | Modified Model | Standard |
|---|---|---|---|
| $\chi^2/df$ | 1.989 | 1.632 | <3.0 |
| RMSEA | 0.089 | 0.071 | <0.08 |
| CFI | 0.841 | 0.914 | >0.9 |
| TLI | 0.824 | 0.900 | >0.9 |
| PNFI | 0.660 | 0.696 | >0.5 |
| PGFI | 0.622 | 0.641 | >0.5 |

**Note:** $N$ = 126; *p*-value = 0.00.

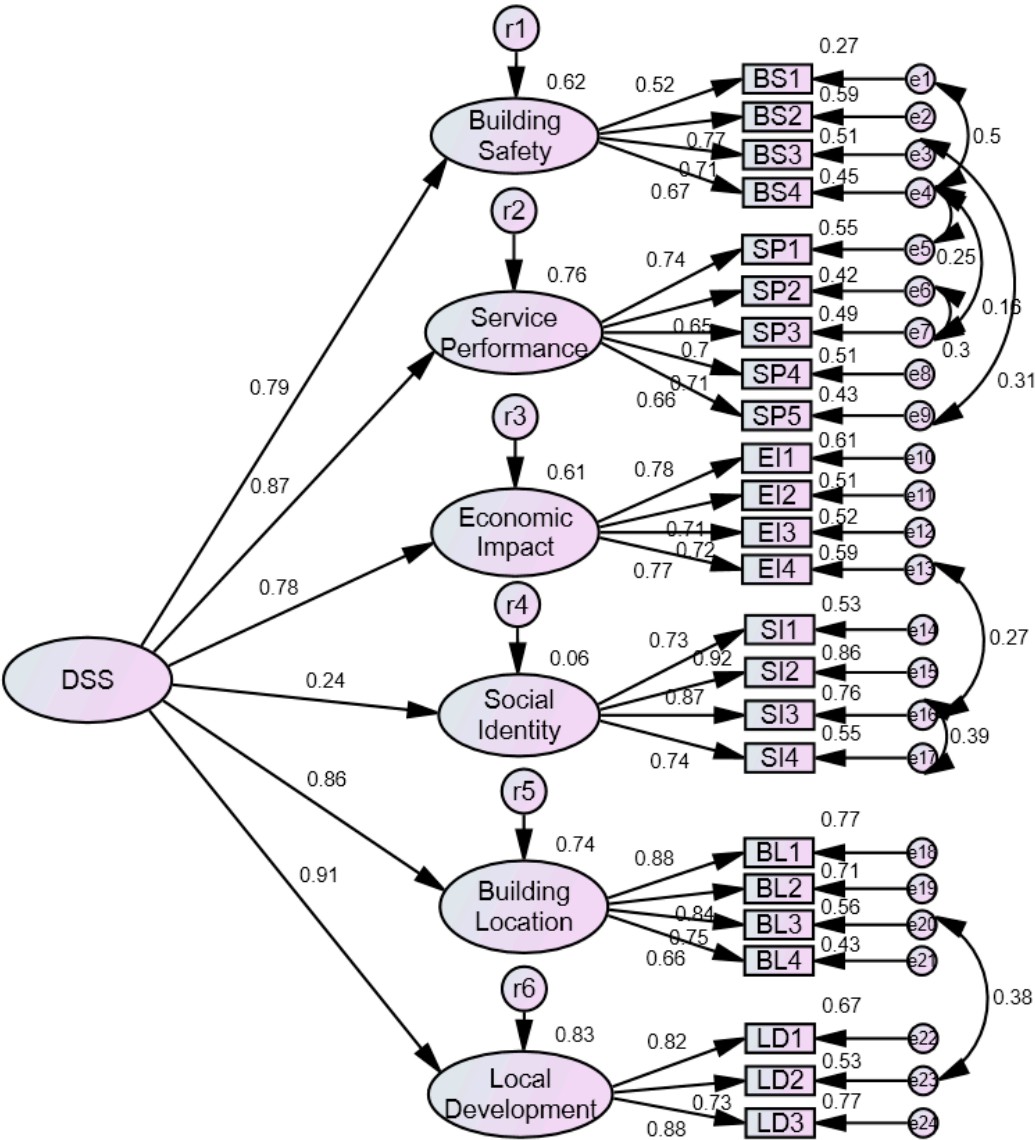

**Figure 1.** Modified second-order CFA model (measurement model).

*4.6. Reliability and Validity Tests*

Having assessed the overall fit of the model, its reliability and validity were tested. Reliability relates to the consistency of a set of observed indicators, while validity relates to the underlying cause of the indicators' covariation [62]. The validity of each path in the CFA model was evaluated using a standardized lambda coefficient, while reliability was evaluated using a squared multiple correlation coefficient ($R^2$) [53]. Both the validity coefficient and $R^2$ value range from 0 to 1. As the validity coefficient approaches 1, the indicator reveals a higher level in representing the construct of interest. Similarly, as the $R^2$ value approaches 1, greater is the variability in each indicator accounted for by the unobserved variable. The lower acceptable limit for the validity coefficient is 0.5. However, there is no consensus on the lower limit for accepting $R^2$ [53,66]. For the observed variables, the *t* value associated with each loading is significant at the 0.001 level, and all the standardized loadings (standardized lambda coefficients) range from 0.523 to 0.925. This exceeds the minimum threshold. The *t* value associated with each latent variable corresponding to the higher-order factor is significant at the 0.05 level, and all the standardized loadings are higher than 0.7, with the singular exception of the loading between the DSS and Social Identity, which stands at 0.24. While this loading is evidently

low, the latent variable of Social Identity was nevertheless retained as it figures prominently in many substantial previous studies as an indispensable dimension in determining urban renewal policy.

Furthermore, in assessing the reliability of the scale, the composite reliability (CR) was calculated [67]. It has been recommended in earlier studies that the universally agreed-upon minimum acceptable threshold for the CR is between 0.6 and 0.7 [56,66]. A similar measure is the average variance extracted (AVE), which is complementary to the composite reliability and which expresses the amount of variance computed for the latent variables [68]. The lower limit for an acceptable level of AVE is 0.5, but values nearer the threshold of 0.50 (>0.45) are accepted for newly developed scales [67]. Applying these tests to the data, we found that both the CR and the AVE scores for all six constructs exceed the recommend level (see Table 9). Specifically, the CR and AVE scores, given respectively for each construct, were Building Safety (0.767, 0.542), Service Performance (0.822, 0.481), Economic Impact (0.763, 0.447), Social Identity (0.872, 0.633), Building Location (0.851, 0.591), and Local Development (0.849, 0.654). This exhaustive and complete set of tests verified the proposed modified model as offering a valid best-fit description of the collected data. Indeed, the developed DSS, containing 24 critical indicators, grouped under six factors, provides a robust model for determining the viability of retaining or relinquishing buildings under consideration for demolition.

**Table 9.** Results of reliability and validity tests of the modified CFA model.

| Survey Item | DSS | BS | SP | EB | CV | BL | LD | $R^2$ | CR | AVE |
|---|---|---|---|---|---|---|---|---|---|---|
| **Building Safety** | **0.788** | | | | | | | **0.621** | **0.767** | **0.542** |
| BS1 Structural safety | | 0.523 | | | | | | 0.274 | | |
| BS2 Environmental safety | | 0.767 | | | | | | 0.588 | | |
| BS3 Natural disaster | | 0.712 | | | | | | 0.507 | | |
| BS4 Fire safety | | 0.674 | | | | | | 0.454 | | |
| **Service Performance** | **0.870** | | | | | | | **0.757** | **0.822** | **0.481** |
| SP1 Usage convenience | | | 0.744 | | | | | 0.554 | | |
| SP2 Unit interior | | | 0.651 | | | | | 0.424 | | |
| SP3 Indoor comfort | | | 0.697 | | | | | 0.486 | | |
| SP4 Amenities condition | | | 0.715 | | | | | 0.511 | | |
| SP5 Life-cycle consumption of resources and energy | | | 0.657 | | | | | 0.432 | | |
| **Economic Impact** | **0.779** | | | | | | | **0.607** | **0.763** | **0.447** |
| EB1 Rate of return on investment | | | | 0.776 | | | | 0.602 | | |
| EB2 Land expectation value | | | | 0.706 | | | | 0.498 | | |
| EB3 Building value | | | | 0.718 | | | | 0.516 | | |
| EB4 Market demand on existing building | | | | 0.770 | | | | 0.593 | | |
| **Social Identity** | **0.236** | | | | | | | **0.056** | **0.872** | **0.633** |
| SI1 Architecture art | | | | | 0.726 | | | 0.527 | | |
| SI2 Historic cityscape | | | | | 0.925 | | | 0.856 | | |
| SI3 Continuity of history | | | | | 0.871 | | | 0.759 | | |
| SI4 Architecture features | | | | | 0.742 | | | 0.551 | | |
| **Building Location** | **0.863** | | | | | | | **0.745** | **0.851** | **0.591** |
| BL1 Availability of public facilities | | | | | | 0.875 | | 0.766 | | |
| BL2 Commercial location | | | | | | 0.837 | | 0.701 | | |
| BL3 Traffic availability | | | | | | 0.751 | | 0.564 | | |
| BL4 Consistency of regional planning | | | | | | 0.658 | | 0.433 | | |
| **Local Development** | **0.911** | | | | | | | **0.830** | **0.849** | **0.654** |
| LD1 Rigid demand of urban building space | | | | | | | 0.816 | 0.666 | | |
| LD2 Industrial structure adjustment | | | | | | | 0.726 | 0.527 | | |
| LD3 Infrastructural investment | | | | | | | 0.877 | 0.769 | | |

## 5. Discussions

### 5.1. Comparison and Evaluation of the Critical Indicators in the DSS

The results demonstrate that the newly developed DSS is grounded in economic, social (e.g., Social Identity), environmental, and institutional considerations, and therefore meets the requirement of informing sustainable urban renewal [2]. Moreover, it was found that the proposed DSS is richer and more comprehensive in its range of parameters than those found in previous research [2]. As a decision support system, both static (e.g., structural safety) and dynamic (e.g., rate of return on investment) indicators are accommodated, allowing the DSS to resolve multiple objectives. Moreover, the DSS consists of internal indicators and external indicators at both the neighborhood level and local level. Furthermore, qualitative indicators (e.g., Social Identity) play as significant a contributory role in the decision-making of building demolition as do quantitative indicators.

In reviewing the 24 indicators, it is clear that there are some appreciable differences between the results reported here relative to previous research findings. Indeed, certain indicators are reported here for the first time, having not been captured in previous studies. Specifically, these are the fixed demand of urban building space and industrial structure adjustment. On the other hand, certain other indicators were found to be less important than previously reported. Specifically, these are building structural safety and social identity indicators [1]. Most surprisingly, the indicator of households' wishes, which relates to community stakeholder involvement in the demolition process, was rejected in the final decision support system. This indicator has intuitive appeal and has been reported in earlier interview studies as a determinant in the decision-making relating to building demolition [2]. Nevertheless, it remains prudent to accommodate stakeholders' attitudes, while community involvement should be encouraged during the decision-making processes.

### 5.2. Locational and Local Conditions of Buildings

Of all the parameters, Local Development is the most critical dimension, being most positively correlated with decision-making in the demolition of existing buildings in urban renewal projects (see Table 7). To be clear, this means that decision-makers lay the greatest emphasis on local development as they move forward in implementing urban renewal projects. Given that earlier studies rate land use, housing policy, and infrastructure as necessary considerations for sustainable urban renewal, this finding corroborates those previous studies [2]. Furthermore, given the continuing rise in urban populations combined with changing housing preferences, the redevelopment of existing buildings is invariably bound to meet the goal of housing more people and at a higher standard [69]. Such an objective can be regarded as efficient land reuse. However, property-oriented urban renewal projects have led to a dominant objective of merely flooding renewal areas with an endless series of cost-effective high-rise apartments, with little regard for the negative impacts such projects have on the sustainable development of a city [70]. This is particularly the case in city centers where land is at a premium and the cost of real estate is high. Consequently, the fixed demand of urban building space is an important indicator, limiting the excessive property-oriented redevelopment of existing buildings. Moreover, changes in social infrastructure investment and industrial structure contribute to the adjustment of urban spatial structure and land redevelopment [9]. This is verified in this study, with both these indicators revealed to be essential considerations in achieving sustainable urban renewal [71].

The factors of Building Location and Service Performance share a similar significance in the decision support system. The indicators underlying the Building Location dimension refer to the service conditions of the existing buildings, such as availability of public facilities and transportation availability. However, while the indicators of Service Performance operate at the building level and neighborhood level, the indicators for Building Location operate at the local level.

### 5.3. Physical Conditions of the Building and Environment

The factor of Service Performance plays a more important role in the decision-making relating to building demolition than does Building Safety. This is because the primary driver for most developers to upgrade or renovate their properties is to improve the service quality and comfort level. This imperative consistently leads to the demolition of otherwise structurally sound buildings when improving the service performance of existing buildings cannot be easily achieved through their renovation. Indeed, most examples displaying the adaptive reuse of existing buildings occur with industrial or commercial buildings. This is because such buildings offer interior and exterior spaces with much greater flexibility than do residential buildings. A well-publicized example is the renovation of the former industrial "798 complex", northeast of Beijing, which is now home to a thriving artist community and is an international tourist attraction. Nevertheless, Building Safety still has a meaningful effect on the demolition decision-making support system. Every year, millions of existing buildings continue to be demolished because of poor safety conditions [72]. Among the five indicators of Service Performance, the $R^2$ values of both the usage convenience and amenities condition exceed 0.5, which means these two indicators have relatively high significance to Service Performance. The former is related to the physical condition of the building, while the latter pertains to the characteristics of the neighborhood. In terms of Building Safety, the factor loadings of environmental safety and natural safety are higher than those of structural safety and fire safety. However, environmental safety has received insufficient emphasis in the continuing practice of urban planning in China. As a typical case, 304 buildings built close to a hazardous chemicals warehouse were damaged during explosions which occurred in Tianjin in August 2015. Consequently, environmental impact should be taken into account as a necessary consideration in making decisions on building demolition.

### 5.4. Social Identity and Economic Impact

The factor Social Identity is much less significant in building demolition decision-making than expected, with the lowest $R^2$ value (0.056) of the six dimensions (see Table 7). Culture is one necessary component of urban development, and it can help to improve social cohesion and community well-being as well as shape the city's image [73]. The ancient residential area of Kuanzai Xiangzi, in Chengdu, is a prime example. However, it seems that the role of Social Identity in urban renewal projects remains less of a concern among planners and developers in China. As reported in previous studies, and in accordance with some of the best practices of urban renewal in developed countries [74], urban renewal planners and decision-makers ought to recognize and facilitate the significant contributions that urban culture and heritage can make to urban development projects. In this study, the four social identity indicators can be divided in two categories: architectural value (architectural art and architectural features) and historical value (historic cityscape and continuity of history). Historical value has a higher level of influence to the dimension of Social Identity as the $R^2$ values of both these indicators exceed 0.7 (see Table 7). Furthermore, it is evident that the historic cityscape has the most significant impact on Social Identity. This result implies that heritage buildings and landmark buildings are an important consideration in urban renewal and ought not to be ignored, though stakeholders appear to pay social identity less attention than it deserves.

Economic Impact used to be the primary consideration for the evaluation of urban renewal projects in China [75]. However, the results of this study reveal that economic considerations should be moderated by a range of other factors which also feature importantly in assessing urban renewal projects, generally, and in the determining of the benefits of demolishing existing buildings to make way for new construction initiatives, specifically (see Table 7). Overall, these findings provide a meaningful instrument for evaluating the merits of building demolition across a wide range of relevant indicators. The proposed DSS provides guidance and offers a means of navigating the decision process to achieve sustainable urban renewal.

## 6. Conclusions

The demolition of vast swathes of the existing urban fabric of cities has been an ongoing feature of urban renewal in China for some decades now. In this process, much has been lost, including a great stock of buildings that could have been refurbished or otherwise used. Much of the social identity that affords social cohesion has gone as well. In the continuing fervor to renew the urban landscape, it is imperative that a robust decision support system be available that can better inform decisions and prevent or reduce the unnecessary destruction of buildings.

This study developed such a decision support system. It consists of 24 indictors selected from literature review and from expert interviews via EFA and CFA. From the perspective of sustainable urban renewal, the system indicators cover economic, social, institutional, and environmental aspects. In comparison with previous related studies, the system not only considers the inherent characteristics of the individual building, but also takes into account the neighborhood characteristics and local context. It is therefore far more comprehensive than earlier support initiatives.

By employing structural equation modelling, 24 identified indicators were grouped into six clusters: service performance, economic impact, social identity, local development, building location and building safety. Based on the results of the confirmatory factor analysis, the most important dimension to emerge was local development (loading = 0.991). The second most important factor revealed that service performance (loading = 0.870) has become more important than safety. This result indicates that the majority of existing buildings that come to be demolished are in fact structurally sound but come to be demolished simply because it may be more economically rational to build again rather than refit existing buildings to the higher service quality desired. Moreover, the results also show that building location (loading = 0.863), building safety (loading = 0.788), economic impact (loading = 779), and social identity (loading = 0.236) all bear a significant impact on the decision-making process when considering the demolition of buildings in renewal projects.

Nevertheless, while social identity rated far less significantly than did other indicators, it ought to remain an important dimension of the decision process. The low standing given by respondents to the surveys for social identity may simply mean that the issue has yet to arouse the necessary awareness it deserves in the minds of planners and developers active in delivering the many urban renewal projects underway in China.

**Author Contributions:** Conceptualization, K.X. and G.Q.S.; methodology, K.X. and G.L.; software, K.X.; validation, K.X.; formal analysis, K.X.; investigation, K.X.; resources, G.Q.S. and G.L.; data curation, K.X.; writing—original draft preparation, K.X.; writing—review and editing, I.M.; visualization, K.X.; supervision, G.Q.S. and G.L.; funding acquisition, G.Q.S. and K.X.

**Funding:** This research was funded by Natural Science Foundation of Zhejiang Province of China (grant number LQ18G030011) and Social Science Foundation of Zhejiang Province of China (grant number 19NDQN334YB).

**Acknowledgments:** The research presented in this paper was funded by Natural Science Foundation of Zhejiang Province of China (grant number LQ18G030011), Social Science Foundation of Zhejiang Province of China (grant number 19NDQN334YB), and the project of "Developing an Integrated Collaborative Platform for Sustainable Urban Renewal: A Pilot Study". The authors would like to express their sincere gratitude for the support of ZNSF, ZSSF, and Hong Kong Polytechnic University.

**Conflicts of Interest:** The authors declare no conflict of interest.

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
