# Peer review of "Demolition of Existing Buildings in Urban Renewal Projects: A Decision Support System in the China Context"

_sustainability, doi:10.3390/su11020491_

Round 1
Reviewer 1 Report
The paper presents a novel decision support system (DSS) for building demolition, from the perspective of sustainable urban renewal. The authors present both a strong theoretical background and potential practical use. The theoretical part is supported by a wide literature review followed by other scientific-based methods such as statistically analyzed stakeholder surveys, exploratory factor analysis (EFA) and confirmatory factor analysis (CFA).
The structure of the paper is clear and the quality of the writing is high. The conclusion is based on clearly presented results supported by comprehensive discussion.
The proposed new DSS presents an interesting and science-based new method to support decision makers on the way to new sustainable urban development processes.
I would like to congratulate the authors for their excellent and high-quality research work.
Author Response
Dear reviewer,
Thank you very much for your kind comments. Please find the attached revised manuscript in the system.
Kind regards,
Kexi Xu
Reviewer 2 Report
Authors deal with an interesting topic of delay factors in decision-making in the process of buildings demolition in urban renewal projects. Although the topic is very interesting, there are several insufficiencies that need to be improved. These insufficiencies can be summed up in highlighting the papers goal in the abstract and introduction as well as research design. Rather than having a large table (e.g. Table 2), that is hard to follow, try showing the generic research process instead. Try showing data from Table 2 in text together with comments.
Author Response
Dear reviewer,
Thank you very much for your comments. Please kindly find the attached file as all the comments have been responded in it.
Kind regards,
Kexi Xu

Reviewer 3 Report
Dear authors,
The topic is completely relevant according to emergent situation about no recognized decision support system for evaluating the merits of building demolition.
Decision support system DSS for building demolition is well defined taking into consideration both: building characteristics and local conditions for sustainable urban renewal.
The literature review is well done.
The methodology is valid based on an exploratory factor analysis (EFA) and confirmatory factor analysis (CFA) for indentifying twenty-four critical indicators covering qualitative and quantitative factors.
The indicators classified under the parameter 3) cultural value , I considered that would be more appropriate to underline the group of indicators under the parameter social identity.
Suggested parameters and changes:
1) service performance √
2) economic benefit - economic impact
3) cultural value - social identity
4) local development √
5) building location - building site
6) building safety √
raw 200: Did You mean EDA for Exploratory Data Analysis instead of EFA? or Exploratory Factor Analysis ?
raw 201: The same doubt about Confirmatory Data Analysis (CFA)?
The factor loadings of environmental safety and natural safety are higher than those of structural safety and fire safety is logical issue according to huge environmental impact by building demolition.
The Chapter 5 (Discussions) could be summarized better and with more precision.
Thank You.
Best regards.
Author Response
Dear reviewer,
Thank you very much for your kind comments. Please find the attached file as all the comments have been responded in it.
Kind regards,
Kexi Xu

Reviewer 4 Report
Title:
ok
Abstract:
It would be necessary to include exact values or percentages of the most significant evidences or results extracted from this investigation.
Keywords:
It is recommended to sort the words from the most specific to the most general
Introduction:
It is considered adequate; however, the study of the knowledge of the problem of this research requires to include the most recognized technical aspects of sustainability assessment and scientific value, such as LCA. Authors are asked to include this aspect. Since without it, knowledge of the subject of sustainable urban renewal is limited. This requirement is considered mandatory.
It is necessary not only to expose the different surveys used by other researchers, it is necessary to specify:
What are they used for? What real application have they had? What impact on urbanization have they achieved?
Authors are asked to review this work:
https://www.sciencedirect.com/science/article/pii/S0360132318300830?via%3Dihub
https://www.sciencedirect.com/science/article/pii/S2210670718311946
Methodology:
All particularities, methods, procedures, variables, etc. that allow other researchers to replicate the work must be defined in this section. Authors are asked to improve this section until reaching an appropriate degree of specificity.
It is necessary to include all the information of the computer programs used (brand, model, year, country, etc.).
Figure 1: it is necessary to improve the quality of the format, to avoid that the numbers coincide with the lines.
Conclusions:
The authors are requested to complete the information including quantitative data of the results identified in this investigation.
Bibliography:
Ok
Author Response

(The authors gave the same response as above.)

Round 2
Reviewer 4 Report
The authors have resolved the comments requested.
The work can be published.
Congratulations